# Resolving DNA Damage: Epigenetic Regulation of DNA Repair

**DOI:** 10.3390/molecules25112496

**Published:** 2020-05-27

**Authors:** Panagiotis Karakaidos, Dimitris Karagiannis, Theodoros Rampias

**Affiliations:** 1Biomedical Research Foundation of the Academy of Athens, 11527 Athens, Greece; pkarak@bioacademy.gr; 2Department of Genetics and Development, Columbia University Medical Center, New York, NY 10032, USA; dk3039@cumc.columbia.com

**Keywords:** DNA damage, DNA repair, epigenomics, chromatin remodeling

## Abstract

Epigenetic research has rapidly evolved into a dynamic field of genome biology. Chromatin regulation has been proved to be an essential aspect for all genomic processes, including DNA repair. Chromatin structure is modified by enzymes and factors that deposit, erase, and interact with epigenetic marks such as DNA and histone modifications, as well as by complexes that remodel nucleosomes. In this review we discuss recent advances on how the chromatin state is modulated during this multi-step process of damage recognition, signaling, and repair. Moreover, we examine how chromatin is regulated when different pathways of DNA repair are utilized. Furthermore, we review additional modes of regulation of DNA repair, such as through the role of global and localized chromatin states in maintaining expression of DNA repair genes, as well as through the activity of epigenetic enzymes on non-nucleosome substrates. Finally, we discuss current and future applications of the mechanistic interplays between chromatin regulation and DNA repair in the context cancer treatment.

## 1. Introduction

The concept of epigenetics has significantly evolved since it was introduced more than 70 years ago when the embryologist Conrad Waddington introduced the term in order to explain how genotypes give rise to phenotypes during development [1]. Epigenetics is a very extensive field and despite how much this area has evolved, the definition of epigenetics is still under intense discussion as investigators ascribe different definitions to the term. On the mid 1990s, the discovery of DNA methylation as a mechanism for gene expression silencing and the identification of imprinted genes, provided a direct link between mitotically or meiotically heritable changes in gene function [2]. In this direction, Arthur Riggs and colleagues defined epigenetics as “the study of mitotically and/or meiotically heritable changes in gene function that cannot be explained by changes in DNA sequence.

The elucidation of nucleosome structure and recent advances on chromatin biology along with research on posttranslational histone modifications, eventually led to broader definitions of epigenomics in order to include gene activity states that can also be transient or occur in non-dividing cells and therefore are not mitotically inherited [3]. Recently, it was proposed that epigenetics could be considered as the study of both transient and inheritable gene expression changes that are unrelated to DNA sequence changes. Moreover, the term “memigenetics” was introduced to specifically describe inherited chromatin activity states [4]. According to this concept, epigenetic regulation includes chromatin modifications (DNA methylation, histone modifications), nucleosome remodeling, alteration of chromatin architecture, expression of non-coding RNAs, and the network of chromatin-modifying and -binding factors that drive or regulate these changes. Some epigenetic signatures and associated transcriptional states such as DNA methylation are stably inherited across many cell divisions and/or generations, while others are transient and can be erased or reversed. In this manuscript we provide insights into the epigenetic processes that participate in transcription and affect the expression of DNA repair components and into the epigenetic processes that are restricted to structural functions and mediate the recognition of DNA damage by DNA repair factors, the accessibility of repair complexes to the sites of damage, the choice of DNA repair pathway and finally, the fidelity of repair process.

Methylation of cytosine residues of CpG dinucleotides on DNA is one of the best characterized epigenetic modifications. This modification plays a critical role in the formation and maintenance of heterochromatin, while its presence in promoter regions with high CpG content correlates with transcriptional repression [5]. A large number of covalent modifications to histone proteins (methylation, phosphorylation, and acetylation) have been described to either promote an open chromatin state that facilitates transcription or a more condensed state that facilitates heterochromatin formation and gene silencing [6]. Chromatin remodeling includes alterations in nucleosome positioning driven by ATP-dependent chromatin remodeling complexes [7] as well as incorporation of histone variants such as H3.3 and H2A.Z that promote specific epigenetic states and play crucial roles in chromosome segregation, transcriptional regulation, and DNA repair [8]. Covalent modifications, nucleosome remodeling, and histone variants function together in order to regulate chromatin processes that require dynamic alteration of DNA structure such as transcription, repair, and replication. Notably, chromatin regulating enzymes have been shown to modify non-histone substrates as well, resulting in regulation of cellular processes such as cell cycle control, differentiation, growth, metabolism, apoptosis, and senescence. In this review we discuss the extended interplay between chromatin regulation and DNA repair, as well as current efforts to apply this knowledge in the clinic.

Large-scale cancer mutation mapping and identification by next generation sequencing in the past few years revealed high frequency of mutations in genes that regulate epigenetic modifications in DNA and histones or chromatin-remodeling processes [9]. These mutations were found to affect chromatin regulator enzymes that catalyze modifications (writers), enzymes that modify or revert the modification (editors/erasers), and enzymes that interact with epigenetic modifications (readers). This type of epigenome deregulations in cancer lead to genome-wide changes in DNA methylation, nucleosome positions, and histone modifications which in turn, affect the local and global chromatin architecture as well as gene expression [10,11,12]. These changes can affect the efficiency of DNA damage response (DDR) and repair and can allow damage to accumulate at much higher rate leading to genomic instability, which is recognized as hallmark of cancer progression [13].

## 2. Types of DNA Damage and DNA Damage Response

### 2.1. Sources of DNA Damage

Cellular DNA is constantly being damaged by harmful chemical modifications driven by oxidation, alkylation, free radicals, ultraviolet and ionizing radiation. Oxidative stress generated by endogenous metabolic processes generates 8-oxo purine monomers such as the 7,8-dihydro-8- oxoguanine (8-oxoG) and other types of oxidized base lesions such as the 8-hydroxy-2-deoxyguanosine (OHdG), and 5-hydroxyuracil. The accumulation of 8-oxoG and OHdG lesions has a severe mutagenic effect because these bases mismatch with adenine during DNA replication [14]. Damaging chemical modifications on DNA can also lead to replication fork stalling during the S-phase of cell cycle and further increase DNA damage [15]. Other types of DNA damage include single stand DNA breaks (SSBs), double strand DNA breaks (DSBs), adducts, intrastrand and interstrand cross-links, and insertion/deletion mismatches. Ionizing radiation (IR) and ultra violet (UV) light represent exogenous sources of DNA damage. Ionizing radiation produce a wide variety of DNA lesions. Among them, DSBs are considered the most cytotoxic and deleterious ones for genomic integrity since, if they are improperly repaired, they can lead to loss of genetic material, chromosomal duplications or translocations, and carcinogenesis [16]. The predominant UV light induced chemical DNA alteration is dimerization of adjacent pyrimidine bases (usually thymine bases) leading to cyclobutane dimers [17]. Other environmental agents that induce DNA damage are alkylating and crosslinking agents [18].

Deregulated origin firing or licensing during DNA replication can also lead to DNA damage. Collision of replication forks has been shown to cause severe lesions, including DSBs and re-replication of DNA [19]. Entering S phase with a reduced number of active origins is associated with long replication tracks and replication fork stalling as well as incomplete replication and chromosome breakages during mitosis [20]. On the other hand, uncontrolled origin activity and increased origin firing can exhaust deoxynucleoside triphosphates (dNTPs) and replication protein A (RPA) levels, causing replication fork collapse and DNA damage [21]. Dysfunctional origin activity and replication stress is induced by oncogene activity and is considered a hallmark of cancer [22].

### 2.2. DNA Repair Is a Multi-Step Process

DNA repair, cell cycle checkpoints, and cell death pathways collectively respond to DNA damage in order to reduce its deleterious consequences to genomic integrity. The DNA repair machinery includes a complex network of sensors, transducers, and effectors that coordinate the repair of DNA damage and ensures DNA replication fidelity. Sensor proteins recognize alterations on the DNA structure such as nicks, gaps, DSBs, and replication lesions. Signal transducers are enzymes that initiate signaling cascades to adjacent nucleoprotein complexes promoting the activation of the cell cycle checkpoints and DNA repair pathways. Effectors repair the DNA damage and inhibit cell cycle progression. In mammalian cells, at least nine distinct pathways are involved for the repair of different genotoxic lesions: mismatch repair (MMR), base excision repair (BER), nucleotide excision repair (NER), translesion synthesis (TLS), homologous recombination (HR), non-homologous end joining (NHEJ), alternative end joining (alt-EJ), Fanconi anemia (FA), and O6-methylguanine DNA methyltransferase (MGMT). Despite the fact that these pathways repair different types of lesions in distinct cell cycle phases, they form a precisely regulated network of multifunctional DNA repair hubs. DSBs are mainly repaired through NHEJ and HR. Whereas NHEJ simply ligates the double helix with no or minimal processing, HR requires the formation of a 3′ ssDNA through the processing of DSB ends by specific nucleases and relies on the recognition and pairing of this 3′ssDNA tails with an intact homologous sequence [23].

### 2.3. DDR Initiation Requires Recruitment of Sensor Proteins to Damaged Chromatin

A key element of DDR is early detection of damaged DNA, followed by recruitment of specific DNA repair complexes to the damaged sites that initiate the repair signaling. After DNA damage, sensor proteins such as PARP1 or H2AX, and complexes, including the Ku70/80 and MRN (MRE11/RAD50/NBS1) directly recognize the structure of DSBs and SSBs and recruit ATM, ATR, and DNA-PK proteins at the break sites. PARP-1 has long been demonstrated as an important first-line sensor of SSBs [24]. Recent structural studies have shown that two flexibly linked N-terminal zinc finger structural elements of this protein, recognize the extreme deformability of SSBs and induce allosterically the activation of the C-terminal catalytic domain [25]. On the other hand, MRN and Ku70/80 complexes are considered as major sensors for DSBs. MRN complex has the ability to bind free DNA ends by one-dimensional sliding. Dynamic DNA binding within the Rad50 domain allows MRN to slide through homoduplex DNA, whereas DNA end recognition is catalyzed by the Mre11 subunit. In contrast, Ku recognizes free DNA ends exclusively via direct binding [26]. PARP1 catalyzes the formation of linear chains of ADP-ribose residues (PAR chains) that form a platform to recruit DNA repair proteins via their PAR-binding domains at the sites of DNA damage [27,28]. Like PARP1, the Ku complex, consisting of Ku70 and Ku80 subunits, functions as a damage sensor, but promotes DNA repair through a different pathway. Binding of the Ku complex to DSBs recruits and activates the DNA-PK catalytic subunit, which promotes NHEJ [29], contrary to PARP1 that promotes alt-EJ [30]. Thus, the antagonism between PARP1 and the Ku complex at DSBs may play an important role in determining repair choices. In addition to PARP1 and Ku proteins, the MRN complex, acting as a DNA sensor protein, can prime the activation of DNA repair cascades via recruitment of ATM to the region near DNA lesions [31]. ATM is the main kinase responsible for the phosphorylation of H2AX, which is one of the earliest steps for the recruitment of additional DDR factors [32,33]. DNA repair pathways are highly linked, and depend on the cellular pathways that modulate locally the chromatin structure at damaged sites in order to ensure accessibility of damage sensors, signal transducers and repair proteins.

Mispaired bases that arise from polymerase misincorporation errors, destabilize the DNA duplex forming a flexible kink on the DNA structure that is recognized by MutS complex in mismatch repair (MMR). This recognition results in the formation of a stable ATP-bound MutS sliding clamp that encircles the DNA double helix and can efficiently diffuse along the DNA backbone. The MutS sliding clamp then recruit MutL onto the mismatched DNA, forming a MutS-MutL complex for the efficient repair [34].

In mammalian nucleotide excision repair (NER), a wide variety of DNA lesions such as ultraviolet light (UV)-induced pyrimidine dimers, as well as intra-strand crosslinks and bulky adducts that are induced by electrophilic chemicals, are recognized by the XPC-RAD23B repair complex. The local thermodynamic destabilization and structural distortion of the double helix by loss of Watson–Crick base pairing, is ubiquitous across diverse NER lesions and provide the basis for their recognition by XPC-RAD23B. Recent studies have demonstrated that this complex discriminates between damaged and non-damaged sites by sensing differences in local deformability on the undamaged strand of the lesion [35,36]. Moreover, the crystal structure of Rad4, the yeast XPC orthologue, bound to DNA containing pyrimidine dimers, revealed that Rad4 recognizes lesions that locally destabilize the Watson–Crick double helix in a manner that permits the insertion of the Rad4 b-hairpin through the DNA duplex and the flipping-out of the two damaged base pairs [37].

The base excision repair (BER) machinery is specialized to fix single base damage in the form of small chemical modifications such as oxidation, alkylation, and deamination that do not usually cause DNA helix distortions. For instance, 8-oxoG is very stable when paired with cytosine in duplex DNA, and the 8-oxoG-containing DNA is not thermodynamically destabilized [38]. DNA glycosylases play a key role in the recognition of such DNA lesions. All glycosylases share a common mode of action for damage recognition; they rotate bases out of the DNA helix into a catalytic pocket, the architecture of which permits a sensitive detection of altered chemical base structure. If the damaged base fits to active site configuration, cleavage of the glycosidic bond will occur, resulting in the generation of apurinic/apyrimidinic sites. Therefore, the detection process requires the scanning of the chemical surface of single bases across DNA [39]. A schematic representation of the early steps of DNA damage response in MMR, BER, and NER pathways is shown in Figure 1.

## 3. Chromatin Regulation during DNA Repair

### 3.1. Epigenetic Driven Chromatin Remodeling Is One of the Earliest Responses to DNA Damage

The first step in DSB repair by HR is the processing of the ends by specific repair nucleases, however, the dense multilevel packing of chromatin poses a challenge for the DNA repair machinery, which requires access to DNA lesions. Early biochemical studies provided evidence that chromatin structure is remodeled in the presence of DSBs and that chromatin is more sensitive to nucleases [40,41,42]. Recent methodological and technological advancements revealed, also, that long range clustering of DSBs occurs on active genes [43] and suggest that DSB bearing chromatin becomes extensively remodeled within the boundaries of the containing TAD (topologically associated domain), bringing the nucleosomes into spatial proximity to facilitate access of repair factors and γH2AX spreading [44]. Similarly, UV irradiation was found to induce a relaxation to chromatin within the entire nucleus [45,46].

This damage-induced chromatin remodeling facilitates also the access of repair proteins to DNA lesions. Prior studies using localized laser micro-irradiation and GFP-tagged H2B histone revealed that chromatin is rapidly and locally relaxed following DSB induction by laser micro-irradiation and then enters a re-compaction phase, necessary for full DDR activation [47,48]. According to these studies, the initial de-condensation of irradiated chromatin was found to be ATM independent and extremely rapid, occurring within seconds after damage. Notably, de-condensation kinetics were found to be similar to the kinetics of PARP1 driven poly ADP-ribosylation, suggesting that chromatin relaxation is associated with PARP1 activity in early DDR. Indeed, inhibition of PARylation abolished chromatin relaxation at DNA damage sites indicating that chromatin relaxation is PARylation dependent [49].

The recent development of CRISPR/Cas9 tools to induce multiple DSBs at known positions in the genome, combined with ChIP-seq and/or super-resolution microscopy, magnified our potential to study the nature and function of chromatin at DSBs. Work from many laboratories in epigenetics have now revealed that these structural alterations are driven by dynamic changes of chromatin modifications and nucleosome positioning after DNA damage, supporting a model where chromatin unfolding and spatial expansion of chromatin states that occur during DNA damage is mediated by specific epigenetic and chromatin remodeling factors.

PARylation has been involved in the recruitment and regulation of several chromatin-remodeling enzymes, whose ATP-dependent activity could promote chromatin relaxation. It is considered that the nucleosome remodeler Arc1 (also known as CHD1L), which contains an ADP-ribose–binding domain, induces an initial relaxation phase [49], which in turn promotes the recruitment of additional remodelers such as chromodomain helicase DNA-binding proteins 3/4 (CHD3 and CHD4 respectively) [50,51], SMARCA5 and CHD2 [52,53]. The activity of chromatin remodelers in combination with the addition of a large number of negatively charged PAR chains onto chromatin contribute to the rapid relaxation of the chromatin structure at DNA damage sites.

Besides PARP1 driven ADP-ribosylation of histones, post-translational modifications of histone tails at damage sites also have a crucial role on chromatin remodeling at early stages of DDR. These modifications can modulate inter-nucleosome contacts, altering higher-order chromatin organization or provide binding sites for recruitment of DDR transducers and effectors. Several groups have demonstrated that rapid H4 acetylation (H4K16ac) is catalyzed on damaged sites and extends for many kilobases away from the break, following the spreading pattern of γH2AX accumulation [54,55]. The N-terminal tail of histone H4 interacts with the acidic patch on the surface of H2A-H2B dimers of adjacent nucleosomes. Disruption of this interaction by acetylation of histone H4 at DSBs promotes chromatin unpacking and formation of open, relaxed, chromatin structures [56]. Acetylation of lysine 16 of histone 4 (H4K16) at DSBs is carried out by the histone acetyltransferase TIP60 and its inactivation is sufficient to block chromatin decompaction during DNA damage [57].

### 3.2. Regulation of Chromatin During Repair of DSBs in Heterochromatin

Nuclear centromeric, pericentromeric regions, telomeres, and repetitive elements comprise the constitutive heterochromatin, while developmentally silenced genes constitute the facultative heterochromatin. Both constitutive and facultative heterochromatins represent highly compacted chromatin, therefore recognition of DNA damage and initiation of early DDRin these regions require a step of chromatin unpacking. Furthermore, the presence of repetitive DNA elements within heterochromatin can initiate aberrant homologous recombination events during repair, such as sister chromatid exchanges or inter-chromosomal recombination, leading to deletions, duplications, translocations, and formation of dicentric or acentric chromosomes. Despite this risk, homologous recombination (HR) is a primary pathway for heterochromatin repair [58]. Thus, chromatin unpacking of heterochromatin during DSB repair should be precisely regulated in order to prevent inappropriate recombination events.

Initiation of HR repair and chromatin unpacking at sites of heterochromatic DSBs requires the phosphorylation of HP1-interacting protein KAP-1 by ATM. This phosphorylation reduces the strength of KAP-1 interaction with damaged heterochromatin, and promotes the release of the chromatin modifier CHD3.1 (CHD3 isoform1). The release of CHD3.1 in turn, drives chromatin relaxation which provides access to DNA repair complexes [59,60].

While the exact molecular mechanism that links damaged DNA sensing and chromatin unpacking at sites of heterochromatic DSBs is unknown, it might involve the histone acetyltransferase activity of Tip60. Upon DNA damage, MRN complex recruits Tip60 at heterochromatic DSBs. This binding is promoted also by the ability of Tip60 to interact with histone H3 trimethylated on lysine 9 (H3K9me3). Tip60-H3K9me3 interaction, activates Tip60′s acetyltransferase activity which leads to damage signaling kinase ATM acetylation and activation [61]. ATM activation in turn, promotes KAP-1 phosphorylation and chromatin unpacking as mentioned above.

Notably, recent studies in Drosophila and mouse cells have shown a relocation of heterochromatin DSBs for the completion of HR repair. This relocation ensures safe and precise HR, while preventing aberrant recombination, by isolating the DSBs and their homologous templates away from ectopic sequences before strand invasion [62]. In Drosophila cells, SUMOylation of HR components after resection has a crucial role in this mechanism as it generates a temporary block to Rad51 recruitment inside the heterochromatin domain to prevent ectopic recombination [63]. Subsequently, damaged sites relocate to anchoring points at the nuclear periphery, in a process that requires the checkpoint kinase ATR [58] and histone demethylase activity by the lysine demethylase 4A (KDM4A) [64]. In contrast, DSBs in mouse cells relocate to the periphery of heterochromatin domains where they are stabilized by the RAD51/BRCA2 [65].

### 3.3. Epigenetic Factors and DNA Repair Choice between HR and NHEJ

Soon after initiation of DDR, the stage is set for repair of the DSB by competing factors that promote NHEJ or HR repair, and less frequently alt-EJ. More specifically, factors compete for initiation of DNA end resection, the first step of the HR repair pathway. The chromatin landscape has been shown to influence this competition depending on its structure and histone marks that are present. For example, SWI/SNF and Ino80 complexes have been shown to compete for chromatin remodeling and levels of H2A.Z exchange, the latter of which promotes HR repair by H2A.Z eviction, DNA end-resection, and presynaptic filament formation [66,67,68]. Moreover, histone marks associated with open chromatin such as H4 acetylation and H3K36me3 have also been associated with HR repair, since chromatin relaxation favors end-resection [69,70]. Conversely, in regions of heterochromatin, NHEJ is favored because of compact chromatin and repetitive elements which hinder HR [65]. Importantly, there is also spatial distinction between HR repair and NHEJ in the nucleus, since when HR takes place in heterochromatin regions the DSB site is extruded as described previously [58]. Finally, demethylation of histone 4 at lysine 20 (H4K20me2) promotes NHEJ by 53BP1 recruitment, reportedly as a way to ensure utilization of this pathway in pre-replicative chromatin [71,72,73]. These interactions have important implications in contexts where the choice of DNA repair pathways is critical, such as cancer treatment.

### 3.4. Epigenetic Regulation of Mismatch Repair

Mismatch repair is required for the detection and correction of single-nucleotide mismatches that might escape proofreading during replication. In addition, MMR is essential for correcting small insertions and deletions that are generated when replication complexes move across repetitive sequences (microsatellites). The MMR process includes recognition of the mispairing, followed by DNA endonuclease nicking of the daughter strand, excision of the daughter strand, and gap-filling by a DNA polymerase. Recognition of replication errors by MMR is mediated by MSH2/MSH6 (MutSα) and MSH2/MSH3 (MutSβ) heterodimers [74]. Binding of MutS complexes to the DNA lesion facilitates the recruitment of the MutL complex (heterodimer MLH1/PMS2) which initiates DNA repair by a single-stranded nick [75]. Other components of the MMR machinery include the exonuclease1 (EXO1) and polymerases with proofreading activity, such as polymerase ε and δ (POLE and POLD). Loss of MMR activity due to mutational inactivation or transcriptional silencing of any of its key players is associated with tumor development and microsatellite instability (MSI). At the genomic level, MMR-deficient tumors (MMRd) are characterized by a hypermutator-phenotype and accumulate large numbers of small insertions and deletions (indels). MMRd tumors represent approximately 20% of human cancers but are unequally clustered across different tumor types. High prevalence is observed in endometrial (~30%), gastric (~20%), and colorectal cancers (~15%), while the proportion of MMR in other tumor types is lower.

There is increasing evidence that epigenetic and remodeling factors play a crucial role in efficient MMR activity. The epigenetic histone mark H3K36me3 seems to be important for recruitment of MMR complexes during DNA replication [76]. Functional experiments have demonstrated that human Msh6 protein harbors a PWWP motif that mediates binding to the H3K36me histone mark during the S-phase of the cell cycle [77]. Moreover, cells lacking the H3K36 tri-methyltransferase SETD2 display characteristics of MMR-deficient cells such as microsatellite instability and high mutation burden [78]. Recent studies have also revealed that SMARCAD1, an SNF-2 family nucleosome remodeler with ATPase activity is an important MMR factor that facilitates nucleosome exclusion from post-replication regions with mismatch base pair. SMARCAD1 interacts with Msh2 protein and facilitates the excision step in EXO1-dependent MMR [79].

Although MMR deficiency has been studied for decades, large amounts of exome sequencing data, now available from The Cancer Genome Atlas (TCGA), revealed that MMRd tumors exhibit a high frequency of frameshift mutations in important chromatin remodelers and histone modifiers genes such as ARID1A, ARID1B, ARID2, ARID4A, EP300, CREBBP, KMT2C, and KMT2D suggesting that MMRd tumorigenesis is associated with epigenetic deregulation [80].

### 3.5. The Role of Chromatin Remodelers in BER and NER

In eukaryotes, base excision repair (BER) is the major pathway for the repair of DNA alkylation and oxidation. Chemically damaged nucleotides, such as lesions of 8-oxoguanine (8-oxoG), activate the BER pathway. Initially, a glycosylase, cleaves the glycosidic bond that links the lesion to the sugar-phosphate backbone and generates an abasic site. Eleven glycosylases have been identified so far in humans and are categorized based on their structure [81]. Subsequently, the apurinic or apyrimidinic sites are bound by the endonuclease APE1, which cleaves the DNA backbone on the 5′ side of the abasic deoxyribose phosphate, creating a nick [82]. The polymerase synthesis step of BER employs either repair polymerase Pol β which adds a single nucleotide, or one of the processive polymerases Pol δ or Pol ε, adding up to 13 nucleotides to the 3′ hydroxyl group of the nucleotide 5′ of the nick [83]. The remaining deoxyribose phosphate is removed by the deoxyribophosphate lyase activity (dRPase activity) of Pol β, whereas the 5′ stretch of nucleotides, when added by Pol δ or Pol ε, is cleaved by the flap endonuclease FEN-1 [84]. The final step of BER is ligation of the nicked strand by DNA ligase IIIα in a complex with its partner protein XRCC1. For execution of this multi-step repair process it is necessary that DNA is accessible to the enzymatic activity of all BER components. Numerous studies have demonstrated an inverse correlation between the level of chromatin compaction and BER activity [85]. Several in vitro studies also indicate that chromatin remodeling activity by members of SWI/SNF subfamily and ISW1/ISW2 proteins is sufficient to facilitate the glycosylase APE-1 and the polymerase synthesis step during BER [86,87,88]. It is considered that chromatin remodelers provide accessibility to BER repair proteins by either remodeling, or combined remodeling and sliding mechanisms.

Nucleotide excision repair (NER) is the major pathway for the repair of bulky DNA lesions caused by UV, environmental mutagens, and cancer chemotherapeutic drugs. Two distinct DNA damage recognition pathways can activate NER. Global genome NER (GG-NER) is activated by helix distortions associated with DNA lesions anywhere in the genome. The main damage sensor in GG-NER is the XPC-RAD23B- CETN2 protein complex. Transcription-coupled NER (TC-NER) is activated by stalled RNA Pol II during transcript elongation by a lesion in the template strand [89]. Similar to the BER pathway, numerous in vitro NER assays have shown that the packed nucleosome structure can be a barrier to efficient NER function. In vitro studies with reconstituted mononucleosomes, purified SWI/SNF complexes were found to increase the accessibility of damaged DNA and stimulate NER repair [90,91]. However, it is not clear yet whether SWI/SNF complexes are involved in early NER steps facilitating XPF recruitment and lesion detection, or are recruited by XPF and promote the binding of late NER factors XPG and PCNA. The latter is supported by experiments showing that knockdown of chromatin remodelers BRG1, BRM, and ARID1A/B have no effect on XPC recruitment but can impair the recruitment of late NER factors ERCC1 and XPA [92,93]. Given the broad role of SWI/SNIF complexes in NER and BER repair pathways and the high incidence of mutations in family members across different cancer types, the exploitation of SWI/SNF deficiency induced susceptibilities is crucial for the development of efficient and precise therapies for SWI/SNF-mutated cancers.

Collectively, epigenetic regulation has a crucial role in all major DNA repair pathways. A schematic synopsis of the most common ones is presented in Figure 2.

## 4. Epigenetic Regulation of DNA Repair Factors Though Transcription

### 4.1. Epigenetic Regulation of DNA Repair Component Expression

A vital aspect of efficient cellular response to DNA damage is transcriptional regulation of DNA repair proteins and related factors. Gene expression of DNA repair genes is regulated by master transcription factors, as well as by epigenetic mechanisms. Gene expression regulation of DNA repair components has been studied primarily under conditions of genotoxic stress. DNA damage, such as DSBs, triggers the activation of transcription factors such as p53, BRCA1, NF-κB, and AP-1. The tumor suppressor protein p53 is stabilized by phosphorylation at multiple sites by multiple DDR kinases, including ATM and CHK1/2 [94,95]. Subsequently, p53 leads to transcriptional activation of multiple DNA repair genes of NER, BER, MMR, and NEHJ pathways [96,97,98,99]. Another target of ATM and ATR is BRCA1 [100], which when phosphorylated has been shown to promote transcriptional activation of DNA repair genes [101] by forming transcriptional complexes with multiple partners, including p53 [102]. Furthermore, the NF-κΒ transcription factor is activated during genotoxic stress through the ATM/NEMO axis [103], and has been reported to drive transcription of the DNA repair proteins BRCA2, REV3, and MGMT [104,105,106]. Lastly, a major output of DDR is activation of the MAPK pathway [107]. MAPK signaling leads to the activation of AP-1, which regulates the expression of many DNA repair proteins, including BER, NER, and MMR genes [108,109,110,111,112]. Importantly, all the above-mentioned master transcription factors drive broad transcriptional programs with significant cross-talk, which facilitates induction of a robust cellular response to DNA damage.

A central aspect of basal transcriptional regulation of DNA repair genes is the epigenetic state of the locus. Repressive chromatin marks, such as DNA methylation, have been associated with reduced gene expression and defective DNA repair in a number of genes. For instance, somatic biallelic methylation of the MLH1 promoter is an important mechanism that leads to MLH1 silencing and mismatch repair deficiency in colorectal cancer [113] and endometrial cancer [114]. Associations between MLH1, MSH2 promoter methylation and mismatch repair deficiency in other tumor types have also been reported [115,116,117]. An H3K4me1 enriched enhancer region located 35 kb upstream of the MLH1 transcriptional start site (TSS) has also been implicated in positive regulation of MLH1 expression. Functional studies revealed that CTCF binding and H3K4me1 within this enhancer is critical for enhancer function and MLH1 expression [118].

BRCA1 promoter methylation is also a frequent event in cancer, that is associated with reduced mRNA and protein levels and HR deficiency [119,120]. Methylation of ERCC1, a component of NER, was correlated with reduced expression and chemosensitivity in glioma [121]. Such changes in chromatin state are often stochastic events confined to a genomic locus, and are sustained because they confer advantages in tumor progression, such as genomic instability [13]. We and others have shown that global epigenetic changes in cancer cells can also affect DNA repair through loss of gene expression of essential DNA repair genes. For instance, KMT2C inactivation in solid tumors leads to HR deficiency through loss of KMT2C binding and activating histone modifications in promoters of ATM and BRCA1 genes, leading to reduced expression levels [122]. In the context of cancer treatment, EZH2 overexpression was reported to lead to DNA damage sensitivity through silencing of RAD51 paralog proteins [123], while inhibition of HDAC enzymes has been shown to downregulate HR repair components and sensitize tumors to chemotherapy [124].

### 4.2. Epi-miRs on the Epigenetic Regulation of DDR Components Transcription

It is increasingly clear that the complexity of gene expression regulation is decisively aided by an additional level of control, the one imposed by microRNAs (miRs). More than 1000 miRs regulate the expression of >60% of all human proteins [125]. However, in the past decade, bidirectional interplay between miRs and epigenetics has become evident [126,127,128]. In other words, epigenetic modifications regulate the expression of numerous miRs but, also, several miRs, known as Epi-miRs, regulate, directly or indirectly, the expression of various epigenetic enzymes (mainly DNMTs, HDACs and KDMs). Herein we will focus on the role of Epi-miRs on the epigenetic control of DDR.

Epi-miRs comprise a small fraction (5–10%) of total miRs [127,128]. Of those, mir-29, the first Epi-miR to be discovered, which is known to repress DNMT3A, DNMT3B, and DNMT1 expression, was found to be upregulated upon DNA damage in a p53-dependent manner [129,130]. Oxidative stress also leads to acute accumulation of mir-29 through TGF-β, followed by suppression of Suv4-20h, reduced H4K20me3, DNA repair deficiency, and genomic instability [131]. Moreover, the association between mir-29, expression of DNMT3A, global DNA methylation and DNA damage levels upon irradiation was also confirmed on several mouse brain tissues [132]. In a similar manner, miR-140, through its control over HDAC4 and HDAC7, was shown to participate in the DDR upon exposure to various DNA damaging agents in several tissues [133,134,135].

Similarly, the miR-99 family (miR-99a, -99b, and -100) modulate DDR through suppression of the SWI/SNF chromatin remodeling factor SNF2H/SMARCA5, a component of the ACF1 complex, leading to reduced BRCA1 localization onto damaged sites and reduced rate and overall efficiency of HR and NHEJ repair pathways [136]. Epi-miR’s involvement in the epigenetic regulation of DDR and repair, couples and extends the known strong effect of other miRs on these processes. Collectively, these findings open a new fascinating area of research that can contribute to the efforts to elucidate the highly complex regulatory networks during DDR.

## 5. Nucleosome-Independent Regulation of DDR by Epigenetic Factors

The contribution of epigenetic factors on DDR regulation is not restricted to histone modifications but is now evident through their interaction with numerous non-histone substrates. The prominent role of post translational modifications (PTMs), like phosphorylation and ubiquitination, in DDR and repair processes is well established and extensively reviewed. However, less common modifications like methylation on lysine and arginine residues, imposed by epigenetic enzymes on non-histone proteins, is an emerging field where a new level of regulation in key DDR processes becomes unraveled. Herein we will focus on the contribution of lysine and arginine methylation of non-histone proteins by epigenetic enzymes in DNA damage-related processes (Figure 3).

### 5.1. Non-Histone Lysine Methylation

It is increasingly evident that lysine methylation on DDR proteins influences, positively or negatively, their chromatin binding, enzymatic activity, subcellular localization, and their protein interactions [137,138,139]. The most prominent example of such proteins is p53, a key regulator of numerous cellular processes including DNA damage response and repair. Up to date, four lysine residues, K370, K372, K373, and K382, of p53 have been found mono- or di-methylated [140,141,142,143]. Of these, K370me1, K373me2, and K382me1 by SMYD2, G9a/GLP, and SET8, respectively, have been reported to repress p53 activity. On the contrary, di-methylation of K370 and K382, by unknown so far lysine methyltransferases, promote p53 function facilitating its binding with 53BP1 and PHF20, as well as p53′s recruitment on the promoter of p21 upon DNA damage [141,142,144]. However, the interplay of lysine methylation marks is dynamic. LSD1, the first lysine demethylase (KDM) to be discovered [145], was shown to demethylate K370me2 to the mono-methylation state with subsequent blockade of the p53-53BP1 interaction [144]. Moreover, the presence of K372me1 (catalyzed by SET7) was shown to promote the nuclear localization, stability, and transcriptional activity of p53, as well as to disrupt its interaction with SMYD2. Another report suggested that SET7 ablation in mouse embryonic fibroblasts prevented the p53-TIP60 interaction leading to decreased acetylation of p53 at K317, K370, and K379, underscoring the interplay between PMTs on p53 regulation [146].

Besides p53, the E2F-1 transcription factor is also methylated during DDR. E2F-1 plays a central role in the DNA damage-induced cell death and, in cells lacking p53, its K185 methylation by SET9 suppresses E2F-1 accumulation, thus, impairing its apoptotic function [147]. Interestingly, LSD1 is able to reverse this mark, indicating that cell fate decisions upon DNA damage are further controlled by methylation.

Another complex protein–protein interplay that is controlled through phosphorylation and methylation events that affect processes like cell cycle progression and DDR was evident in the tumor suppressor retinoblastoma (RB) protein. Upon RB phosphorylation, E2F-1 is released and promotes cell cycle progression. Methylation of RB by SMYD2 enhanced its phosphorylation and E2F-1 mediated cell cycle progression [148]. In addition, RB could be dephosphorylated by PPP1R12A when mono-methylated at K442 by SET7 [149]. However, the same report showed that LSD1 could remove this methylation mark, rendering PPP1R12A susceptible to ubiquitination and proteasomal degradation.

Recently, in a large screen for SET7 methylation targets related to DDR, Thandapani et al. identified FEN1 as one. FEN1 was found monomethylated at K377 by SET7 throughout cell cycle. They reported that this mark did not affect the endonuclease activity of FEN1 but it was required for the cellular response to replicative stress [150]. Another target of SET7 is ubiquitin-like with PHD and RING finger domains 1 (UHRF1). UHRF1 methylation at K385 facilitates its interaction with PCNA leading to PCNA polyubiquitination, a necessary event for the initiation of HR repair pathway [151]. Once again, LSD1 is the enzyme that can erase SET7′s mark.

At the onset of DDR, numerous DDR factors become PARylated by PARP1. However, PAR formation and PARP1 recruitment at DNA damage sites following IR is controlled by SET7/9 and SMYD2 through K508 and K528 methylation of PARP1, respectively [152,153]. Additional methylation events that control recruitment of DDR factors include the methylation marks of DNA-PKcs (K1150me3, K2746me2, and K3248me2) and Ku80 (K7me3) that are recognized by HP1β and required for the proper localization of several factors at the damage sites [154].

In cancer cells, heterochromatin relaxation in response to DNA damage is also affected by the methylation of the methyltransferase SUV39H1 at K105 and K123 by SET7/9 [155], highlighting that, during DDR, a functional interplay between epigenetic enzymes with methyltransferase activity is also possible next to the antagonistic events between KMTs and KDMs described earlier.

Interestingly, it is plausible that the extent of regulation of epigenetic factors on DDR proteins may include protein–protein interactions beyond of their enzymatic activities [156]. In a recent report, Alsulami et al. identified a necessary protein–protein interaction between SET1A and RAD18 for the proper recovery after DNA damage [157]. Similarly, protein–protein associations have been identified between SET1 and BOD1L in protecting replication forks [158] and between SETD1A and cyclin K during DNA repair [159].

### 5.2. Non-Histone Arginine Methylation

Methylation of arginine residues are also involved in the complex regulation of DDR and repair processes. So far, four out of nine arginine methyltransferases (PRMTs) have been identified to methylate important DDR factors. PRMT1, the most active enzyme among PRTMs (> 80% of all PRMT activity), di-methylates asymmetrically the DDR factor MRE11 that along with RAD50 and NBS1, form the MRN complex, which serves as a primary sensor of DSBs. Arginine methylation of MRE11′s RGG/RG motif by PRMT1 has been shown to be essential for its nuclease activity, recruitment at the sites with DSBs, exposure of ssDNA, and initiation of HR repair [160,161,162]. Interestingly, in vivo substitution of PRMT1 targeted arginines by lysines of MRE11 impaired also the ATR/CHK1 signaling pathway along with defects in cell cycle checkpoints and gross chromosomal aberrations [163]. Similar to MRE11, 53BP1 was also found methylated at its RGG/RG motif (residues R1406 and R1413) by PRTM1, altering its binding capacity to sites with SSBs or DSBs [164,165]. On the other hand, BRCA1 methylation by PRMT1 alters BRCA1′s efficiency to bind DNA as transcriptional cofactor [166]. PRTM1 and PRTM6 methylate also Polβ at R83, R137, and R152 residues. These methylation modifications affect Polβ’s interaction with PCNA, and alter its polymerase processivity and DNA binding affinity [167,168].

Like PRMT1, PRMT5 is also involved in the regulation of several DDR factors, with the most important being KLF4 and p53. The transcription factor KLF4 plays primal role in the cellular decision of pursuing DNA damage repair. Upon damage, PRTM5 methylate KLF4 at R374, R376, and R377 and, thus, its degradation is prevented. Accumulating KLF4 was found able to activate p21-dependent cell cycle arrest and inhibition of apoptosis [169]. Methylation of p53 by PRTM5 was demonstrated to induce a p53-dependent cell cycle arrest response, while in the absence of PRTM5, a p53-dependent apoptosis was promoted [170,171]. The FEN1 nuclease (participates on both replication and long-patch BER) is also regulated by PRTM5 on up to four arginine residues (primarily R192). Methylation on these specific sites enhances FEN1-PCNA interaction and promotes the nuclear localization of FEN1 at replication sites during S phase or after genotoxic stress [52,54], highlighting the dynamic interplay between PTMs [172,173,174].

As mentioned in the previous section, non-enzymatic functions of epigenetic factors are emerging. Accordingly, Hamard et al. reported that DNA repair efficiency is regulated by PRMT5 through its interference on the alternative splicing of key histone modifying and DNA-repair factors, including EZH2, SETDB1, SUV4-20H2, and TIP60, with the latter being particularly involved in the DNA repair of hematopoietic cells [175]. Collectively, the above reports highlight the importance of lysine/arginine methylation in the regulation of numerous non-histone proteins and cellular decisions implicated in DDR or repair processes (Table 1). Of note, such modifications interfere and/or interact with additional PMTs (phosphorylations, ubiquitinations, acetylations), on the same or partner molecules, magnifying the complexity of the DDR regulatory network. The complete picture seems far from solved, as the field is rapidly expanding and the expectations of identifying arginine demethylating enzymes are high [176].

## 6. Epigenetic Chromatin Regulation and DNA Repair: Synthetic Lethal Interactions and Clinical Applications

As illustrated, chromatin regulation and DNA repair have a complex interplay that we only recently have begun to understand. As a result, there is substantial effort to translate these findings for patient benefit, particularly in cancer. A big portion of cancer treatment options rely on killing cancer cells through induction of DNA damage directly by chemotherapy or irradiation, or indirectly through targeting DNA repair. However, high toxicity and refractory or recurrent disease are frequent, which calls for new treatment options and better patient selection. Epigenomic alterations are thought to play an important role in drug resistance by contributing to gene expression plasticity and tumor heterogeneity [177]. Moreover, the interplay between epigenetic regulation and DNA repair can be exploited to achieve greater therapeutic response, by synergistic and synthetic lethal interactions. In this context, chemical inhibition of epigenetic factors that modulate DDR and drug resistance is a promising and attractive avenue for anticancer therapy (Figure 4).

Chromatin regulation is a complex process, that is often disrupted in various ways within cancer cells. Epigenetic drugs that target components of chromatin regulation, such as inhibitors of DNMTs and HDACs have been proven clinically effective mostly in hematopoietic malignancies that are particularly reliant on epigenetic deregulation of progenitor/stem cells [178]. In solid tumors, broad use of epigenetic drugs has been proven ineffective [179] and only targeted approaches such as use of EZH2 a and IDH inhibitors in selected patients seem promising [180]. A synopsis of current epigenetic drugs used in clinic is shown in Table 2. DNA repair is also frequently disrupted in cancer, and multiple approaches based on targeting DNA repair are currently employed in the clinic as reviewed previously [181].

### 6.1. Epigenetic Inhibitors in Combination with Chemotherapy/Radiotherapy

In the context of chromatin regulation and DNA repair, the most explored therapeutic approach so far is combination of epigenetic inhibitors with chemotherapeutic agents. Such combinations have displayed synergistic effects in pre-clinical models. More specifically, HDAC inhibitors have been shown to inhibit the DDR/HR pathway and cause sensitivity to DNA damage-inducing agents in various cell types [124]. Moreover, HDAC, DNMT, and LSD1 inhibitors were shown to counteract epigenetic resistance mechanisms and restore sensitivity to chemotherapy in solid tumors [182,183,184]. A number of clinical trials were conducted to assess the efficacy of these combinations in treatment of advanced solid tumors with mixed results in terms of patient response and toxicity, likely because of the differences between regimens and cohorts [185,186]. It is likely that more targeted approaches will be more beneficial, as in the context of BRF1 or EGFR bearing mutant non-small-cell lung cancers, where EZH2 inhibition was shown to selectively sensitize these tumors to topoisomerase II inhibition [187].

Following the same rationale, epigenetic inhibitors could also potentiate radiotherapy. Preclinical evidence has shown such synergism between radiotherapy and HDAC inhibitors [188,189], BET inhibitors [190], EZH2 inhibitors [191,192], and DNA methyltransferase inhibitors [193]. Of these, only the combination of HDAC inhibitors and irradiation is under assessment in the clinic and initial results indicate high toxicity and only limited patient benefit. DNA methyltransferase inhibitors, such 5-azacytidine and decitabine, are cytidine analogs that are incorporated in DNA and are potent radiosensitizers in all contexts, so this combination is not viable due to high toxicity. Optimization of regimen and dosage will be key to increasing efficacy of these drug combinations in the clinic.

### 6.2. Epigenetic Inhibitors in Combination with Drugs that Target DNA Repair Components

Another therapeutic approach to exploit this interplay is to utilize synthetic lethal interactions. As mentioned above, HDAC inhibitors have been shown to inhibit expression of HR repair genes and this provides a rationale for combing HDAC and PARP inhibitors to achieve effective tumor killing [194,195]. Such synergism has been observed in pre-clinical models of prostate, breast, and ovarian cancer [196,197,198,199]. The efficacy of combining HDAC and PARP inhibitors is currently under evaluation in the clinic (NCT03742245). Similar effects in the expression of HR components are also observed by BET inhibition [200]. Pre-clinical studies have shown significant synergism between BET and PARP inhibitors in multiple cell types, including breast and ovarian cancer [200,201,202,203], which is also under assessment in the clinic (NCT03991469). Combination of PARP and DNA methyltransferase inhibitors has also shown synergistic activity in AML and breast cancer cells [204]. PARP1 and DNMT1 were shown to interact during repair and simultaneous inhibition led to increased PARP1 trapping and DNA damage. A clinical trial is investigating the efficacy of this combination in AML patients (NCT02878785). All the above combinations allow administration of low doses of each drug, which is particularly important for HDAC and DNMT inhibitors, since high toxicity has been a limiting factor to their application. Optimal patient selection and regimen will be crucial to the success of these and future trials.

### 6.3. Targeting DNA Repair in Tumors with Epigenetic Alterations

Another way to take advantage of the interplay between DNA repair and the epigenome is to target DNA repair components in cancer cells with specific epigenomic alterations. An example of this is H3K36me3 loss which occurs in tumors with either mutations in SETD2, the methyltransferase that deposits this mark, or mutations in histone H3 that inhibit the generation of this modification (e.g., H3K36me3), as well as in tumors that overexpress the demethylases KDM4A and KDM4B [205,206]. These events are frequent in various cancer types, including renal cell carcinoma, lung cancer and glioma [207], and have been associated with poor prognosis [208]. H3K36me3 has been implicated in many DNA repair pathways including HR, NHEJ, and MMR [209]. Pfister et al. identified a dependency of tumors with low H3K36me3 to cell cycle checkpoints, rendering them sensitive to WEE1, CHK, and ATR inhibition [210]. This discovery led to the initiation of a clinical trial assessing the use of the WEE1 inhibitor adavosertib in SETD2-deficient solid tumors (NCT03284385). Based on the role of H3K36me3 in other aspects of DNA repair, it would be interesting to examine other potential synthetic lethal interactions in H3K36me3-low tumors such as PARP inhibition.

Another component of chromatin regulation that has been shown to be actively involved in DNA repair pathways is the subunit of the SWI/SNF complex, ARID1A. In solid tumors, this epigenetic factor is frequently mutated and its inactivation has been linked with aggressive disease [211,212]. ARID1A-deficient tumors were shown to have cell cycle defects due to its role in DNA damage response and cell cycle checkpoint regulation [213,214]. Consequently, these tumors were found to be sensitive to PARP and ATR inhibitors. A number of clinical trials are currently investigating the efficacy of targeting ARID1A deficient tumors with these inhibitors in patients (NCT04065269, NCT03207347, NCT04042831).

## 7. Concluding Remarks

Mammalian cells have evolved a complex network of pathways in order to utilize, protect, and preserve their genetic information. This includes pathways dedicated in the maintenance and regulation of chromatin structures, as well as pathways that recognize and repair genomic lesions. As described above, accurate restoration of DNA damage relies on the direct or indirect activity of chromatin regulators. Specifically, DNA repair components depend on extensive alterations in chromatin state in order to access and repair DNA lesions. Moreover, the activity of chromatin regulators is essential for sustained expression and post-translational regulation of these components. As more evidence accumulates and more interactions are uncovered, we gain better understanding of these processes. Given the extensive involvement of chromatin regulation and DNA repair in tumor development, progression, and treatment, comprehension of the complex interplays between them will be crucial for discovery of effective therapeutic approaches. Specifically, there is need for more effective epigenetic and DNA repair drugs, in terms of target specificity and pharmacokinetics. Additionally, therapeutic approaches that rely on synthetic lethal interactions require efficient patient selection based on genetic or phenotypic properties, such as DNA repair proficiency and chromatin state, which is still only partially implemented in cancer therapy. In conclusion, it is an exciting time for both basic researchers exploring the mechanistic interplays between epigenetics and DNA repair, as well as for clinicians that develop new therapeutic approaches.

## Figures and Tables

**Figure 1 molecules-25-02496-f001:**
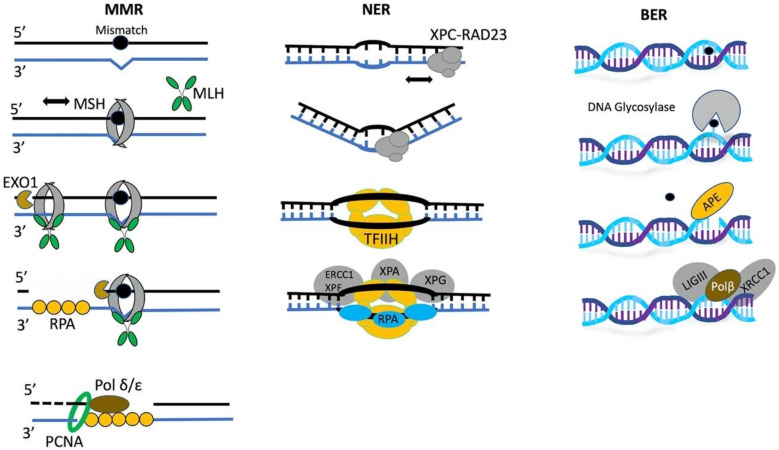
Early steps of DNA damage response that include DNA damage recognition and recruitment of additional repair factors in mismatch repair (MMR), nucleotide excision repair (NER), and base excision repair (BER) pathways.

**Figure 2 molecules-25-02496-f002:**
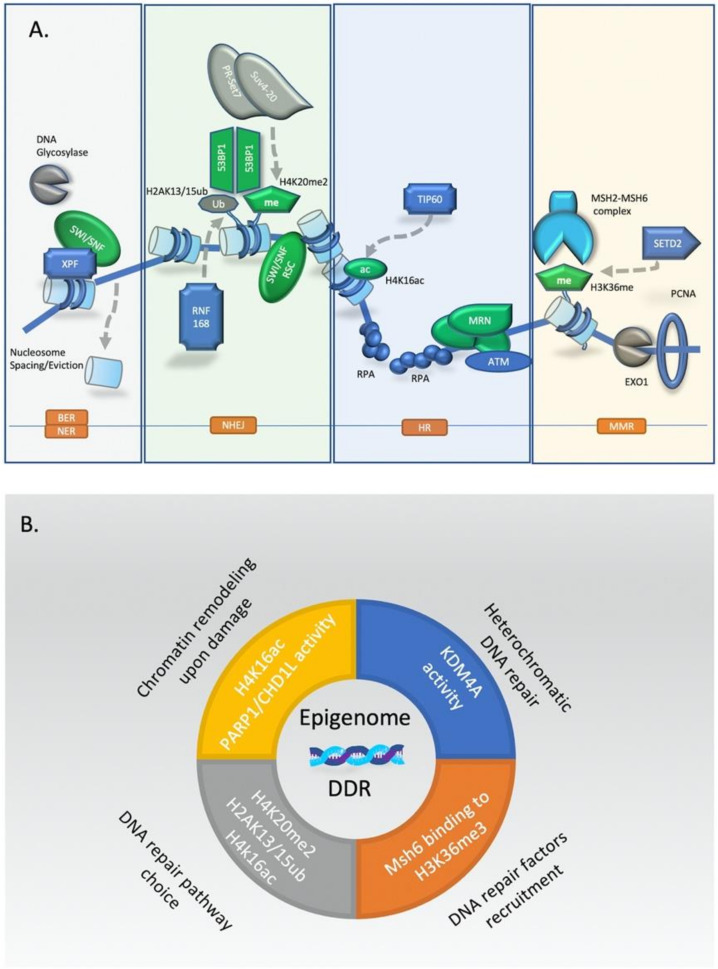
Epigenetic-driven chromatin modifications and DNA repair (**A**) Epigenetic regulation of key DNA repair processes with emphasis in chromatin modifications. (**B**) Multilayer epigenetic regulation upon DNA damage response (DDR).

**Figure 3 molecules-25-02496-f003:**
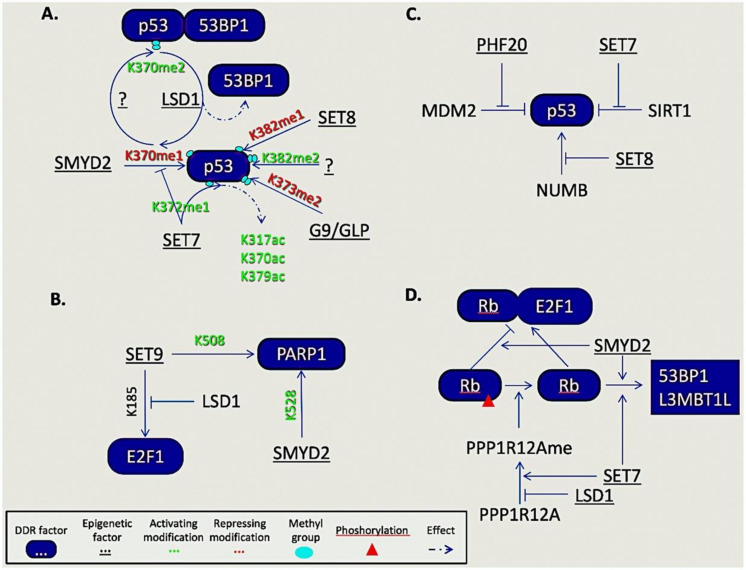
Regulation of key non-histone DDR components by epigenetic factors. (**A**) Direct lysine methylation of p53. (**B**) Indirect regulation of p53 through methylation by epigenetic factors. (**C**) Examples of non-histone epigenetic regulation of DDR components. Of note the opposing roles of SET9 and LSD1 on E2F1 methylation. (**D**) Regulation of Rb-E2F1 axis by epigenetic factors.

**Figure 4 molecules-25-02496-f004:**
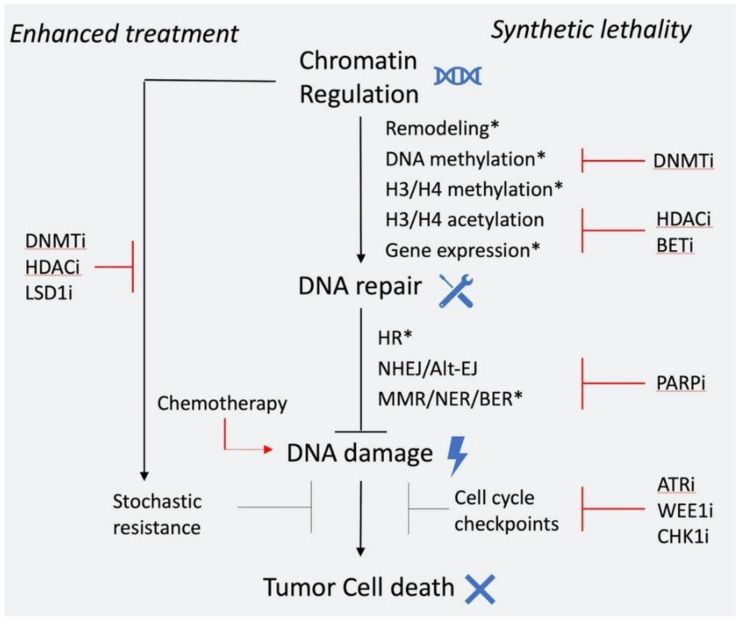
Targeting multiple steps of DNA damage resolution leads to efficient cell killing. Chromatin regulation is necessary for efficient DNA repair. These pathways are often defective in cancer (asterisks *) and can be targeted using specific inhibitors. As a result, synthetic lethal interactions can be exploited with multiple ways, which can be very advantageous in the clinical setting, where changes in treatment are required.

**Table 1 molecules-25-02496-t001:** List of methylated non-histone substrates of epigenetic factors involved in DDR.

	DDR Protein	Methylated Residue	Epigenetic Enzyme	Effect
**LYSINE METHYLATION**	**TP53**	K370me1	SMYD2	repression
		K373me2	G9/GLP	repression
		K382me1	SET8	repression
		K370me2-->me1	LSD1	repression
		K370me2		activation
		K372me1	SET7	activation (promotes p53 acetylation)
		K382me2		activation
	**E2F1**	K185	SET9 LSD1	suppresses E2F1 accumulation
	**NUMB**	K158 K136	SET8	p53 degradation
	**Rb**		SMYD2	enhances Rb phosphorylation
		K810	SET7	docking site for 53BP1
		K860	SMYD2	docking site for L3MBTL1
	**SITR1**		SET7	prevents SIRT1-p53 interaction, p53 activation
	**PPP1R12A**	K442me1	SET7	dephosphorylates Rb
		K442me1-->K442	LSD1	unable to dephosphorylate Rb
	**PCNA**	K248	SET8	activation
	**MDC1**	K45	JMJD1C	activation
	**FEN1**	K377	SET7	activation
	**UHRF1**	K385me1	SET7	PCNA polyubiquitination
		K385me-->K385	LSD1	prevents UHRF1-PCNA interaction
	**PARP1**	K508 K528	SET7 SMYD2	enhances PARP1 at damage sites
	**DNA-PKcs**	K1150me3		activation
		K2746me2		
		K3248me2		
	**KU80**	K7me3		activation
	**SUV39H1**	K105 K123	SET7	heterochromatin relaxation
**ARGININE METHYLATION**	**MRE11**		PRMT1	activation
	**53BP1**	R1406	PRMT1	activation (enhances binding at damage sites)
		R1413		
	**BRCA1**		PRMT1	activation
	**Polβ**	R83	PRMT1 PRMT6	activation
		R137		
		R154		
	**KLF4**	R374	PRMT5	KLF4 accumulation
		R376		
		R377		
	**TP53**		PRMT5	activation
	**FEN1**	R192	PRMT5	activation

**Table 2 molecules-25-02496-t002:** Representative epigenetic drugs used in clinic.

Inhibitor Type	Representative Drugs	Target	Status	Cancer Type
**HDAC**	Vorinostat	All HDACs	FDA approved	T-cell Lymphoma
	Romidepsin	HDAC1-3	FDA approved	T-cell Lymphoma
	Belinostat	All HDACs	FDA approved	T-cell Lymphoma
	Panobinostat	All HDACs	FDA approved	Refractory multiple myeloma
**BET**	OTX015/MK-8628	BRD2/3/4	phase 1b	NUT midline carcinoma
	I-BET762	BRD2/3/5	phase 1/2	NUT midline carcinoma & hematological cancers
**DNMT**	5-azacitidine	DNMTs	FDA appoved	AML, MDS
	Decitabine	DNMTs	FDA appoved	AML, MDS
**HDM**	tranylcypromine	LSD1	phase 1	AML
**HMT**	tazemetostat	EZH2	phase 1/2	B-cell Lymphoma
	Pinometostat	DOT1L	phase 1	MLL-r Leukemia

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
