# Peer review of "Resolving DNA Damage: Epigenetic Regulation of DNA Repair"

_molecules, 2020, doi:10.3390/molecules25112496_

Round 1
Reviewer 1 Report
Title: Resolving DNA damage: Epigenetic regulation of DNA repair. molecules-803879
Panagiotis Karakaidos, Dimitrios Karagiannis, Theodoros Rampias
This is a review article that covers a very wide variety of topics on epigenetic modifications and DNA damaging response and repair. Authors summarized different types of epigenetic modifications, such as DNA methylation, histone and non-histone modifications, miRNA and chromatin remolding, most types DNA repair pathways, and discussed the effects of epigenetic modifications on DNA damage and repair processes. The potential therapeutic applications of targeting epigenetic factors in cancer treatment were discussed at the end of the article.
This review is very informative for beginners in the related fields. However, it has some issues need to be addressed:
1: Maybe because the article covered too many topics, many statements are overly simplified, and some paragraphs are redundant and some are irrelevant to the topics. For example:
- The introduction starts with the term of “epigenetics” and then “However, this term in chromatin biology refers mainly to…”, suggesting this paper will be focused on epigenetic and chromatin biology in DNA damage and repair. The next sentence about cytosine methylation is also focused on the effects of methylation on chromatin structures. However, the rest part of the article does not always follow this topic. Many sections are about DNA/protein modification and DNA repair activities in general, not about chromatin structures.
- Under the subtitle “Regulation of chromatin during repair of DSBs in heterochromatin”: most parts within lines 172-190 are not about DSB repair and chromatin structures.
- Lines 511-512 and line 530 are talking the same thing.
- Lines 249-261: This paragraph is about methylation down-regulates gene expression, not about chromatin remodeling. But this section is under the title “3. Chromatin regulation during DNA repair”.
- Page 8: most statements in the first 2 paragraphs are about methylation and gene expression and cancer in general, not about DNA damage-induced response in DNA repair process.
- As stated at the very beginning, epigenetics refers to “heritable traits that were not attributable to changes in DNA sequence”. However, some modifications discussed in the paper are not necessarily inheritable alterations. For example, is there any evidence supporting the non-histone lysine methylation (5.1) and arginine methylation (5.2) are inheritable changes, rather than transient and temporary modification? If so, they should be clarified at the beginning of the sections. Also, phosphorylation of γH2AX usually is not considered at “epigenetic modification” since it is transient and not inheritable. Modifications of repair proteins are also not epigenetic.
- Some modifications on DNA and proteins happen before DNA repair and are (early) part of DNA repair processes. But some alterations might just be the by-products of repair activities. Therefore, the literatures need to be interpreted with a little more detail. When the proteins or epigenetic factors were listed, readers may wonder which “sensor” proteins sense the damage and which enzymes pass the signal to the mentioned proteins, to make these epigenetic factors part of the DNA repair activities. For example, in lines 179-193: which protein(s) sense the damage in heterochromatin “before” the condensed chromatin structure was relaxed? Which marks are essential for DNA repair?
5: 2.1 Types of DNA damage. – this section is rather “sources of DNA damage”
6: 2.2 DNA repair is a multi-step process. “… Sensor proteins recognize alterations on the DNA structure such as nicks, gaps, double-strand breaks (DSBs), and replication lesions” --- Author may also want to mention how DNA repair proteins sense the DNA helix distortions caused by damage before they can nail down to the nicks, gaps and adductions. This is also part of the “chromatin structure” which should be the focus of this paper.
Reviewer 2 Report
This review gives a summary of how epigenetics influence DNA repair. The authors take the approach of giving a broad overview of the topic and not focusing on a particular area or pathway. As such, it lacked some depth and felt too dense at spots. However, it was well researched and will serve as a good overview of the topic for researchers new to the field of epigenetics and DNA repair. The authors also pose some interesting questions and connections about how epigenetics influence DNA repair and affects cancer progression and treatment. The section on clinical applications and epigenetic inhibitors was particularly well done. While there are some grammatical errors throughout, the review is well written and properly cited. In my opinion, this review will be of interest to those in the field and researchers going into it. Below are a few minor comments and suggestions to improve the review:
- Some minor English editing is needed throughout the manuscript.
- Section 2 is basically a list of types of damage, repair pathways and the DDR. It was a lot to cram into the small space and did not provide enough detail. Two examples include the title of section 2.2, which states DNA repair if a multi-step process but pathways are only named and steps not included, and in section 2.3 the authors discuss the DDR for DSBs but what about the other pathways? A figure showing the basic pathways/major player involved or focusing on one pathway at time would improve this section. Also, much of the detail on the pathways are included in section 3 so another idea is to incorporate this section into section 3.
- In a similar vein to the comments above, in section 3, creating separate DSB repair from the other forms of repair (i.e. BER, NER, MMR) would help with the flow of the review.
- Line 240-243: Citations are needed for both of these sentences.
- Line 277: Sentence states, “Numerous studies…” but only one study is cited.
- The text in Figure 1A is very small and difficult to see. Increasing either the figure or font size is needed.
- In Figure 2, the green is text is difficult to see with the gray background.
- The text in both tables is very small and hard to see.
- The title of section 4 would be more accurate to include “…transcription of DNA repair factors”
Author Response
"Please see the attachment."

Reviewer 3 Report
In this review, Karakaidos et al. reviews our current understanding of how chromatin is modified during DNA damage and its repair process. Further the review explores the transcriptional regulation during DNA Damage Response (DDR) and also the role of chromatin modifying enzymes on non-nucleosome protein involved in DDR. Finally, the authors describe the clinical applications of the epigenetic regulation of DDR. Overall the review is extensive and well-written. However, I have one major comment:
Large-scale chromatin conformation as measured by 3C based technologies is now a well-studied property that is often considered under the umbrella of epigenetics. Recent studies have shown long-range clustering of DNA double-strand break sites (Aymard F et al. 2017) and also discuss the role of TADs in DNA repair (Arnould C et al, 2020). This level of regulation is only briefly mentioned in the manuscript in its current form.
Author Response
"Please see the attachment."

Round 2
Reviewer 1 Report
Authors addressed most comments and concerns this reviewer had and the revised manuscript was improved dramatically. Although I still don't agree with authors 100% in a few places, but as a review article, I think it is OK to leave these to readers.